# Resilient Multiple Choice Learning: A learned scoring scheme with application to audio scene analysis

**Victor Letzelter**[1,2]  **Mathieu Fontaine**[2]

**Mickael Chen**[1]  **Patrick Pérez**[1]  **Slim Essid**[2]  **Gaël Richard**[2]

[1] valeo.ai, Paris, France
[2] LTCI, Télécom Paris, Institut Polytechnique de Paris, France

## Abstract

We introduce Resilient Multiple Choice Learning (rMCL), an extension of the MCL approach for conditional distribution estimation in regression settings where multiple targets may be sampled for each training input. Multiple Choice Learning is a simple framework to tackle multimodal density estimation, using the Winner-Takes-All (WTA) loss for a set of hypotheses. In regression settings, the existing MCL variants focus on merging the hypotheses, thereby eventually sacrificing the diversity of the predictions. In contrast, our method relies on a novel learned scoring scheme underpinned by a mathematical framework based on Voronoi tessellations of the output space, from which we can derive a probabilistic interpretation. After empirically validating rMCL with experiments on synthetic data, we further assess its merits on the sound source localization task, demonstrating its practical usefulness and the relevance of its interpretation.

## 1   Introduction

Machine learning models are commonly trained to produce, for any given input, a single prediction. In most cases, for instance, when minimizing the empirical risk with quadratic loss, this prediction can be interpreted as the conditional output expectation given the input. However, there are many tasks for which the conditional output distribution can be multimodal, either by the nature of the task or due to various sources of uncertainty. Temporal tracking and forecasting, for instance, are problems of this type [5, 32, 44]. In such cases, the mean might fall in a low-density region of the conditional probability function, and it would be beneficial to predict multiple hypotheses instead [34, 20]. In this context, Multiple Choice Learning (MCL) has emerged as a simple pragmatic solution [17, 29]. Thanks to a network architecture with multiple heads, MCL models can produce multiple hypotheses, one per head. During supervised training, the gradients are only computed for the head that provides the best prediction given the current input sample. This Winner-Takes-All (WTA) training scheme allows each head to specialize in a region of the output space. Accordingly, Rupprecht et al. [34] have proposed a probabilistic interpretation of MCL based on Voronoi tessellations.

On that account, MCL suffers from two significant issues: *hypotheses collapse* and *overconfidence*. *Hypotheses collapse* occurs during training when a head takes the lead, the others being almost never selected under WTA and thus not updated. This leads to a situation where most heads are not trained effectively and produce meaningless outputs during inference. *Overconfidence* can also be observed at inference time when looking at MCL under the lens of its probabilistic interpretation. Hypotheses that correspond to rare events tend to be over-represented, thus not reflecting the true distribution of

37th Conference on Neural Information Processing Systems (NeurIPS 2023).

outputs well. Multiple variants of MCL have been developed to alleviate these issues, in particular for classification or segmentation tasks [29, 27, 42, 13].

In this paper, we are interested in MCL in the context of regression settings, which has been mostly overlooked. More specifically, we propose rMCL, for *resilient Multiple Choice Learning*, a MCL variant with a learned scoring scheme to mitigate overconfidence. Unlike prior work, rMCL is designed to handle more general settings where multiple targets may be sampled for each input during training. Moreover, we propose a probabilistic interpretation of rMCL, and show that rMCL's scoring fixes overconfidence by casting multi-choice regression as conditional distribution estimation. While introduced in the context of the original WTA loss, the proposed scoring approach is also compatible with other variants of WTA. We validate our claims with experiments on synthetic data, and we further evaluate the performance of rMCL on the sound source localization problem. The accompanying code with a rMCL implementation is made available.[1]

## 2    Related Work

**Uncertainty estimation.**  The estimation of machine learning model's uncertainty can be studied from different perspectives depending on the type of ambiguity being considered. The uncertainty can concern either the model (*epistemic*) or the data (*aleatoric*) [7, 12]. Kendall and Gal [22] showed that those types of uncertainty can be combined in deep learning settings, placing both a distribution on the weights and the outputs of the model. Epistemic uncertainty estimation may, for instance, be achieved by variational approximation of the weights posterior [31, 14]. Independent ensembles (IE) of neural networks [18, 11] is another widespread technique for epistemic uncertainty approximation. Several neural networks, trained from different random initializations, provide samples for approximating the weights posterior distribution [26].

**Multiple Choice learning.**  Originally introduced by Guzman-Rivera et al. [17] and adapted to deep learning settings by Lee et al. [28, 29], MCL is a framework in which multiple output heads propose different possible predictions. Since many datasets contain only a single output realization for each input, the heads are trained with a Winner-Takes-All scheme where only the head that made the best prediction is updated. It can be understood as a specific case of dependent ensembles [8, 46, 4, 47] where the members interact with each other during training. Alternatively, [34] have shown that MCL can be adapted successfully for multi-label classification [23].

The Winner-Takes-All training scheme, however, can cause two main issues known as *hypotheses collapse* and *overconfidence* [34, 6, 10, 19, 10, 27, 42]. Most related previous works tackle either or both of those questions. Hypotheses collapse occurs during training when some heads receive little or no updates due to bad initial values under the WTA loss and other hypotheses taking the lead. To fix this issue, Rupprecht et al. [34] proposed a relaxed Winner-Takes-All (RWTA, or $\varepsilon$-WTA) loss that allows the update of non-winner heads, albeit with gradients scaled by a small constant $\varepsilon$. While this approach ensures that every head receives gradients updates, it also tends to level the different heads, which counters the initial goal of WTA and thus needs careful tuning. Finally, one can leverage the evolving Winner-Takes-All loss [32], with the top-$n$ heads getting updated (top-$n$-WTA) instead of only the best one. The authors validate that their approach, with a scheduled decreasing value for $n$, achieves improved conditioning of the outputs by reducing inconsistent hypotheses in the context of future frames prediction.

On the other hand, overconfidence is an issue that can be observed in inference when evaluating a model as a density estimator. In this case, one would want the different outputs to be distributed across the different modes of the true conditional probability of the outputs. Empirical observations [27, 42] show that this is not usually the case, and rare events tend to be overly represented, rendering such a model inadequate for integration within real-world decision-making systems. In this context, Lee et al. [27] proposed *Confident Multiple Choice Learning* (CMCL) for solving overconfidence in a classification setup. Additionally to the WTA loss, CMCL is trained by maximizing the entropy of the class distributions predicted by the non-selected hypotheses. The classifier's final prediction is based on a simple average of the discrete class distributions predicted by each of the heads. Although designed for tackling the overconfidence problem, CMCL reduces the diversity of the hypotheses. On that account, *Versatile Multiple Choice Learning* [42] (vMCL) proposed to address the issue by leveraging a choice network aiming at predicting a score for each hypothesis head. This may be seen

---

[1] https://github.com/Victorletzelter/code-rMCL

as a variant of Mixture-of-Experts approaches [33], in which the choice network is supervised with explicit targets. The final prediction in vMCL is derived by weighting the class distributions from each hypothesis by their respective scores. These works focus on classification tasks and aggregate the hypotheses, thereby losing individual information from each head. In contrast, we address regression tasks and demonstrate how to benefit from the diversity of predictions *without* aggregation.

We propose to revisit the vMCL approach for regression, single-target or multi-target, and to extend accordingly the mathematical interpretation of the WTA proposed by Rupprecht et al. [34].

## 3 rMCL regression and its probabilistic interpretation

After reviewing MCL, this section introduces resilient Multiple Choice Learning, dubbed rMCL. This proposed variant handles multi-target settings thanks to its scoring scheme. We then provide a distribution learning interpretation of rMCL that is relevant to regression tasks, thus completing the MCL probabilistic interpretations of prior works [9, 34].

### 3.1 Fixing the overconfidence issue in Multiple Choice Learning

Let $\mathcal{X} \subseteq \mathbb{R}^d$ and $\mathcal{Y} \subseteq \mathbb{R}^q$ be the input and output spaces in a supervised learning setting, such that the training set $\mathscr{D}$ is composed of samples $(\boldsymbol{x}_s, \boldsymbol{y}_s)$ from an underlying joint distribution $p(\boldsymbol{x}, \boldsymbol{y})$ over $\mathcal{X} \times \mathcal{Y}$.

Multiple choice learning was proposed to address tasks with ambiguous outputs, *i.e.*, for which the ground-truth distribution $p(\boldsymbol{y} \mid \boldsymbol{x})$ is multimodal [17, 29, 10]. Adapted by Lee et al. [29] for deep-learning settings in *Stochastic Multiple Choice Learning* (sMCL), it leverages several models $f_\theta \triangleq (f_\theta^1, \ldots, f_\theta^K) \in \mathcal{F}(\mathcal{X}, \mathcal{Y}^K)$, referred to as $K$ *hypotheses*, trained using the Winner-takes-all (or Oracle) loss. It consists, given an underlying loss function $\ell$ and for each sample $(\boldsymbol{x}_s, \boldsymbol{y}_s)$ in the current batch, of first the computation of

$$\mathcal{L}(f_\theta(\boldsymbol{x}_s), \boldsymbol{y}_s) \triangleq \min_{k \in [\![1, K]\!]} \ell\left(f_\theta^k(\boldsymbol{x}_s), \boldsymbol{y}_s\right) \tag{1}$$

after each forward pass, followed by backpropagation on the *winner* hypothesis, that is, the minimizing one. Assuming now that a set $\boldsymbol{Y}_s$ of targets is available in the training set for input $\boldsymbol{x}_s$, Firman et al. [10] have shown that (1) can be generalized by updating the best hypothesis per target using the meta-loss

$$\mathcal{L}\left(f_\theta\left(\boldsymbol{x}_s\right), \boldsymbol{Y}_s\right) = \sum_{\boldsymbol{y} \in \boldsymbol{Y}_s} \sum_{k=1}^K \mathbf{1}\left(\boldsymbol{y} \in \mathcal{Y}^k\left(\boldsymbol{x}_s\right)\right) \ell\left(f_\theta^k\left(\boldsymbol{x}_s\right), \boldsymbol{y}\right), \tag{2}$$

where

$$\mathcal{Y}^k(\boldsymbol{x}) \triangleq \left\{\boldsymbol{y} \in \mathcal{Y}, \ \ell\left(f_\theta^k(\boldsymbol{x}), \boldsymbol{y}\right) < \ell\left(f_\theta^r(\boldsymbol{x}), \boldsymbol{y}\right), \ \forall r \neq k\right\}. \tag{3}$$

This loss can, however, lead to poor conditioning of the output set of predictions (*hypotheses collapse*), with only one or a few hypotheses being exclusively selected for backpropagation. Furthermore, sMCL is subject to the *overconfidence* problem. Following Theorem 1 of [34], a necessary condition for minimizing the risk

$$\int_{\mathcal{X}} \sum_{k=1}^K \int_{\mathcal{Y}^k(\boldsymbol{x})} \ell\left(f_\theta^k(\boldsymbol{x}), \boldsymbol{y}\right) p(\boldsymbol{x}, \boldsymbol{y}) \mathrm{d}\boldsymbol{y} \mathrm{d}\boldsymbol{x}, \tag{4}$$

is that each prediction $f_\theta^k(\boldsymbol{x})$ coincides with the centroid of $\mathcal{Y}^k(\boldsymbol{x})$ with respect to $p(\boldsymbol{y} \mid \boldsymbol{x})$. We say that the components $\{\mathcal{Y}^k(\boldsymbol{x})\}_k$ form a *centroidal* Voronoi tessellation [9]. If $\ell$ is for instance the $\ell_2$-loss, this condition means that for each non-zero probability cell $k$, $f_\theta^k(\boldsymbol{x})$ amounts to the conditional mean of $p(\boldsymbol{y} \mid \boldsymbol{x})$ within the Voronoi cell $\mathcal{Y}^k(\boldsymbol{x})$. However, this theorem tells us nothing about the predictions in very low probability zones; in such regions, the inference-time predictions $f_\theta^k(\boldsymbol{x})$ will be meaningless. During inference, the sMCL indeed faces a limitation: it cannot solely rely on the predicted hypotheses to identify Voronoi cells in the output space with low probability mass (see Figure 1). This observation leads us to propose *hypothesis-scoring* heads $\gamma_\theta^1, \ldots, \gamma_\theta^K \in \mathcal{F}(\mathcal{X}, [0, 1])$. Those aim to predict, for unseen input $\boldsymbol{x}$, the probability $\mathbb{P}(Y_{\boldsymbol{x}} \in \mathcal{Y}^k(\boldsymbol{x}))$ where $Y_{\boldsymbol{x}} \sim p(\boldsymbol{y} \mid \boldsymbol{x})$ with the aim of mitigating this overconfidence issue.

While MCL variants have often been proposed for classification tasks, we are, to the best of our knowledge, the first to propose to solve this issue for multi-target regression, interpreting the problem as a multimodal conditional distribution estimation.

## 3.2   Resilient Multiple Choice Learning

We consider hereafter a problem of estimation of a multimodal conditional distribution denoted $p(\boldsymbol{y} \,|\, \boldsymbol{x})$ for each input $\boldsymbol{x} \in \mathcal{X}$. In a real-world setting, only one or several samples (the *targets*) drawn from $p(\boldsymbol{y} \,|\, \boldsymbol{x})$ are usually accessible. Sound source localization is a concrete instance of such multimodal prediction for whose $p(\boldsymbol{y} \,|\, \boldsymbol{x}_s)$ represents the sound source position for an input audio clip $\boldsymbol{x}_s \in \mathcal{X}$ at a given time $t$.

In this last example, each target sample in $\boldsymbol{Y}_s$ may represent the location of a *mode* of the ground-truth multimodal distribution. For such multi-output regression tasks, the MCL training of a randomly initialized multi-hypotheses model with scoring functions, $(f_\theta^1, \ldots, f_\theta^K, \gamma_\theta^1, \ldots, \gamma_\theta^K)$ can be adapted as follows.

For each training sample $(\boldsymbol{x}_s, \boldsymbol{Y}_s)$, let

$$\mathcal{K}_+(\boldsymbol{x}_s) \triangleq \left\{ k^+ \in [\![1, K]\!] : \exists \boldsymbol{y} \in \boldsymbol{Y}_s, k^+ \in \operatorname*{argmin}_k \ell(f_\theta^k(\boldsymbol{x}_s), \boldsymbol{y}) \right\} \tag{5}$$

and $\mathcal{K}_-(\boldsymbol{x}_s) \triangleq [\![1, K]\!] - \mathcal{K}_+(\boldsymbol{x}_s)$ be the set of *positive* (or *winner*) and *negative* hypotheses respectively. It is then possible to combine the multi-target WTA loss $\mathcal{L}$ in (2) with a hypothesis scoring loss

$$\mathcal{L}_{\text{scoring}}(\theta) \triangleq -\left( \sum_{k^+ \in \mathcal{K}_+(\boldsymbol{x}_s)} \log \gamma_\theta^{k^+}(\boldsymbol{x}_s) + \sum_{k^- \in \mathcal{K}_-(\boldsymbol{x}_s)} \log \left( 1 - \gamma_\theta^{k^-}(\boldsymbol{x}_s) \right) \right), \tag{6}$$

in a compound loss $\mathcal{L} + \beta \mathcal{L}_{\text{scoring}}$. This novel approach differs from previous MCL variants [42] in its ability to predict multimodal distributions in regression settings and by the introduction of separated scoring branches that are updated based on loss (6), such that the target for the scoring branch $k$ is the probability that hypothesis $k$ is among the winners for that sample.

A resource-efficient implementation of rMCL is achieved by deriving the hypotheses and score heads from a shared representation, with distinct parameters at the final stages of the architecture (*e.g.*, through independent fully connected layers). Designed as such, it is also possible to reduce memory cost by updating only a fraction of score heads associated with the negative hypotheses at each training step, typically with distinct samples $k^- \sim \mathcal{U}(\mathcal{K}_-(\boldsymbol{x}_s))$. This trick could also alleviate the imbalanced binary classification task that the scoring heads face with a large number of hypotheses $|\mathcal{K}_-(\boldsymbol{x}_s)| \gg |\mathcal{K}_+(\boldsymbol{x}_s)|$ (typically, the number of target samples at disposal is small relative to the number of hypotheses $K$). The variants of the WTA (*e.g.*, top-$n$-WTA, $\varepsilon$-WTA, see Sec. 2) are also compatible with rMCL. Inference with the proposed rMCL model is outlined in Algorithm 1.

As highlighted above, the hypothesis output can be interpreted, in the context of risk minimization, as the conditional mean of the Voronoi cell $\mathcal{Y}^k(\boldsymbol{x})$ it defines, providing that the cell has non-zero probability. Furthermore, given (6), the output of the score head $\gamma_\theta^k(\boldsymbol{x})$ can be interpreted as an approximation of the probability of a sample from $p(\boldsymbol{y} \,|\, \boldsymbol{x})$ to belong to this cell.

---

**Algorithm 1** Inference in the rMCL model

---

**Input:** Unlabelled input $\boldsymbol{x} \in \mathcal{X}$. Trained hypotheses and score heads $(f_\theta^1, \ldots, f_\theta^K, \gamma_\theta^1, \ldots, \gamma_\theta^K) \in \mathcal{F}(\mathcal{X}, \mathcal{Y})^K \times \mathcal{F}(\mathcal{X}, [0,1])^K$.
**Output:** Prediction of the output conditional distribution $p(\boldsymbol{y} \,|\, \boldsymbol{x})$.
1: Perform a forward pass by computing $f_\theta^1(\boldsymbol{x}), \ldots, f_\theta^K(\boldsymbol{x}), \gamma_\theta^1(\boldsymbol{x}), \ldots, \gamma_\theta^K(\boldsymbol{x})$.
2: Construct the associated Voronoi components $\mathcal{Y}^k(\boldsymbol{x})$ (3) with $\mathcal{Y} = \cup_{k=1}^K \overline{\mathcal{Y}^k(\boldsymbol{x})}$.
3: Normalize the predicted scores $\gamma_\theta^k(\boldsymbol{x}) \leftarrow \gamma_\theta^k(\boldsymbol{x}) / \sum_{k=1}^K \gamma_\theta^k(\boldsymbol{x})$.
4: If $Y_{\boldsymbol{x}} \sim p(\boldsymbol{y} \,|\, \boldsymbol{x})$, then for $k$ such that $\gamma_\theta^k(\boldsymbol{x}) > 0$, interpret the predictions as estimations of

$$\begin{cases} \gamma_\theta^k(\boldsymbol{x}) = \mathbb{P}(Y_{\boldsymbol{x}} \in \mathcal{Y}^k(\boldsymbol{x})) \\ f_\theta^k(\boldsymbol{x}) = \mathbb{E}[Y_{\boldsymbol{x}} \,|\, Y_{\boldsymbol{x}} \in \mathcal{Y}^k(\boldsymbol{x})] \end{cases} \tag{7}$$

---

### 3.3 Probabilistic interpretation at inference time

Let us consider a trained rMCL model, such as the one described in the previous section. Following the theoretical interpretation of [34] and the motivation of solving the overconfidence issue, as explained in Section 3.1, the output objectives of the model can be summarized in (7). Although it is possible, assuming that those properties are verified, to deduce the following law of total expectation

$$\mathbb{E}[Y_{\boldsymbol{x}}] = \sum_{k=1}^{K} \boldsymbol{\gamma}_{\theta}^{k}(\boldsymbol{x}) f_{\theta}^{k}(\boldsymbol{x}), \tag{8}$$

the above quantity may not be informative enough, especially if the law of $Y_{\boldsymbol{x}} \sim p(\boldsymbol{y} \mid \boldsymbol{x})$ is multimodal. Indeed, to be able to derive a complete probabilistic interpretation, we still need to characterize how rMCL approximates the full conditional density of $Y_{\boldsymbol{x}}$. For that purpose, it is sufficient to fix a law $\pi_k$ for $Y_{\boldsymbol{x}}$ within each Voronoi cell $k$ to accurately represent the predicted distribution in each region of the output space as per the rMCL model, as shown in Proposition 1.

**Proposition 1** (Probabilistic interpretation of rMCL). *With the above notations, let us consider a multi-hypothesis model with properties of* (7). *Let us furthermore assume that*

$$Y_{\boldsymbol{x}}^{k} \sim \pi_k(\cdot \mid \boldsymbol{x}) \tag{9}$$

*i.e., the law $Y_{\boldsymbol{x}}^{k}$ of $Y_{\boldsymbol{x}}$ conditioned by the event $\{Y_{\boldsymbol{x}} \in \mathcal{Y}^{k}(\boldsymbol{x})\}$ is described by $\pi_k$ for each input $\boldsymbol{x} \in \mathcal{X}$ with constraint $\mathbb{E}[Y_{\boldsymbol{x}}^{k}] = f_{\theta}^{k}(\boldsymbol{x})$. Then, for each measurable set $A \subseteq \mathcal{Y}$,*

$$\mathbb{P}(Y_{\boldsymbol{x}} \in A) = \int_{\boldsymbol{y} \in A} \sum_{k=1}^{K} \boldsymbol{\gamma}_{\theta}^{k}(\boldsymbol{x}) \pi_k(\boldsymbol{y} \mid \boldsymbol{x}) \mathrm{d}\boldsymbol{y}. \tag{10}$$

*Proof.* We have

$$
\begin{array}{rcll}
\mathbb{P}(Y_{\boldsymbol{x}} \in A) & = & \sum_{k} \mathbb{P}(Y_{\boldsymbol{x}} \in A \cap \mathcal{Y}^{k}(\boldsymbol{x})) & \text{(by sigma-additivity)} \\
& = & \sum_{k} \mathbb{P}(Y_{\boldsymbol{x}} \in \mathcal{Y}^{k}(\boldsymbol{x})) \mathbb{P}(Y_{\boldsymbol{x}} \in A \mid Y_{\boldsymbol{x}} \in \mathcal{Y}^{k}(\boldsymbol{x})) & \text{(by Bayes' theorem)} \\
& = & \sum_{k} \boldsymbol{\gamma}_{\theta}^{k}(\boldsymbol{x}) \int_{\boldsymbol{y} \in A} \pi_k(\boldsymbol{y} \mid \boldsymbol{x}) \mathrm{d}\boldsymbol{y} & \text{(by (9)).} \quad \square
\end{array}
$$

Following the principle of maximum entropy [38], the uniform distribution inside each cell $Y_{\boldsymbol{x}}^{k} \sim \mathcal{U}(\mathcal{Y}^{k}(\boldsymbol{x}))$ is the least-informative distribution, assuming that the output space $\mathcal{Y}$ is with finite volume and the hypotheses lie in the geometric center of their associated cell. With this prior, the predicted distribution can be interpreted as

$$\hat{p}(\boldsymbol{y} \mid \boldsymbol{x}) = \sum_{k=1}^{K} \boldsymbol{\gamma}_{\theta}^{k}(\boldsymbol{x}) \frac{\mathbf{1}\left(\boldsymbol{y} \in \mathcal{Y}^{k}(\boldsymbol{x})\right)}{\mathcal{V}(\mathcal{Y}^{k}(\boldsymbol{x}))}, \tag{11}$$

where $\mathcal{V}(\mathcal{Y}^{k}(\boldsymbol{x})) \triangleq \int_{\boldsymbol{y} \in \mathcal{Y}^{k}(\boldsymbol{x})} \mathrm{d}\boldsymbol{y}$ is the volume of $\mathcal{Y}^{k}(\boldsymbol{x})$. Similarly, if we model the output distribution as a mixture of Dirac deltas such that in each cell $k$, $\boldsymbol{\gamma}_{\theta}^{k}(\boldsymbol{x}) > 0 \Rightarrow Y_{\boldsymbol{x}}^{k} \sim \delta_{f_{\theta}^{k}(\boldsymbol{x})}$, then

$$\hat{p}(\boldsymbol{y} \mid \boldsymbol{x}) = \sum_{k=1}^{K} \boldsymbol{\gamma}_{\theta}^{k}(\boldsymbol{x}) \delta_{f_{\theta}^{k}(\boldsymbol{x})}(\boldsymbol{y}). \tag{12}$$

### 3.4 Toy example

We build upon the toy example formalized in [34, §4.1] to validate the proposed algorithm and its interpretation. In particular, we seek to illustrate how rMCL handles the overconfidence problem in a regression task. The toy problem involves the prediction of a 2D distribution $p(\boldsymbol{y} \mid x)$ based on scalar input $x \in \mathcal{X}$, where $\mathcal{X} = [0, 1]$. The training dataset is constructed by selecting for each random input $x \sim \mathcal{U}([0, 1])$ one of the four sections $S_1 = [-1, 0) \times [-1, 0)$, $S_2 = [-1, 0) \times [0, 1]$, $S_3 = [0, 1] \times [-1, 0)$, $S_4 = [0, 1] \times [0, 1]$, with probabilities $p(S_1) = p(S_4) = \frac{1-x}{2}, p(S_2) = p(S_3) = \frac{x}{2}$. Whenever a region is selected, a point is then sampled uniformly in this region. To slightly increase the complexity of the dataset, it has been chosen to enable the sampling of two (instead of one) target training samples with probability $q(x)$ for each input $x$. Here, $q$ is a piece-wise affine function with $q(0) = 1$, $q(\frac{1}{2}) = 0$, and $q(1) = 1$.

We compared the behavior of our proposed approach rMCL against sMCL and independent ensembles (IE) using a 3-layer perceptron backbone with 256 hidden units and 20 output hypotheses, and $\ell_2$ as the underlying distance. For rMCL, we used scoring weight $\beta = 1$. In addition to the full-fledged rMCL, we assess a variant denoted 'rMCL*' where a single negative hypothesis is uniformly selected during training. For independent ensembles, we trained with independent random initialization 20 single-hypothesis models, the predictions of which being combined to be comparable to multi-hypothesis models. The training processes were executed until the convergence of the training loss. We report the checkpoints for which the validation loss was the lowest. All networks were optimized using Adam [24]. We show in Figure 1 (top right) test predictions of the classical sMCL compared to those of the rMCL model. In visualizing the predictions where the ground-truth distribution is known, we observe empirically that the *centroidal* tessellation property, $\mathbb{E}[Y^k_{\boldsymbol{x}}] = f^k_\theta(\boldsymbol{x})$, is verified with good approximation (see red points Figure 1, bottom right). We otherwise notice the *overconfidence* problem of the sMCL model in low-density zones when the output distribution is multimodal by looking at the output prediction, *e.g.*, for $x = 0.1$ and $x = 0.9$ in Figure 1 where the sMCL predictions fail in low-density zones. In contrast, rMCL solves this issue, assigning to each Voronoi cell a score that approximates the probability mass of the ground-truth distribution in this zone (see Figure 1, bottom). Furthermore, the independent ensembles (triangles) suffer from a collapse issue around the conditional mean of the ground-truth distribution. This behavior is expected as the minimizer of the continuous formulation of the risk (4) on the whole output space is the conditional expectation of the output given the input.

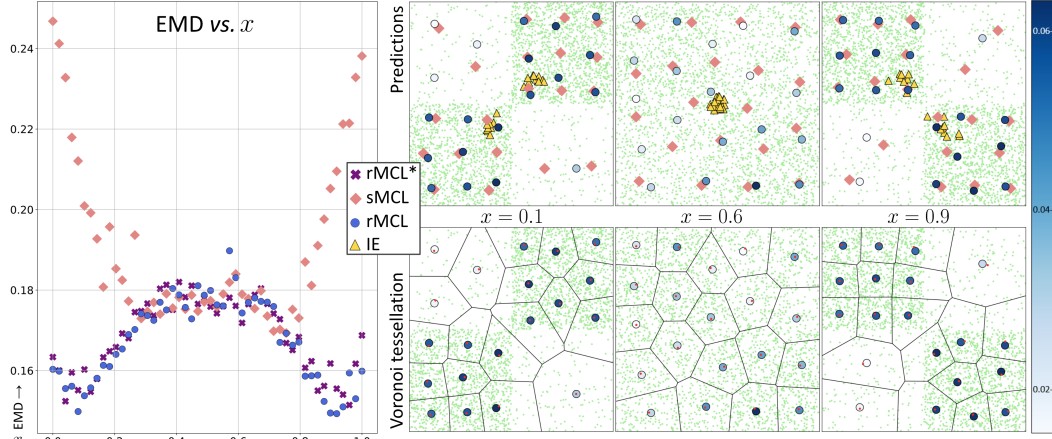

Figure 1: **Comparisons on a 2D regression toy problem**. Comparing sMCL with standard WTA ('sMCL', pink diamonds), proposed rMCL (blue circles), a variant of rMCL with a single negative hypothesis chosen uniformly at training ('rMCL*', purple crosses), and Independent Ensemble ('IE', yellow triangles). (*Left*) Test EMD for different inputs $x$; (*Right*) Inference at $x = 0.1, 0.6, 0.9$ and Voronoi tessellations generated by the hypotheses. Green points are samples from the ground-truth distribution at the corresponding input. Cells conditional mean (red points) matches with the predicted hypotheses. The score predicted in each cell, displayed as color saturation of the blue circles, as per scale on the right, approximates the corresponding proportion of points. At $x = 0.1$ and $0.9$, we can observe how rMCL tackles well the overconfidence issue in the low-density zones.

From a quantitative perspective, as demonstrated in Fig. 1, we can evaluate the effectiveness of rMCL in addressing the *overconfidence* issue compared to the classical sMCL. To ensure a fair comparison with sMCL settings, we have opted for using the Earth Mover's Distance (EMD) metric, which measures the discrepancy between the ground truth and predicted distributions. Predicted distributions are considered mixtures of Dirac deltas as in (12) for the metric computation, with sMCL predictions assigning uniform mode weights. As depicted in Fig. 1, rMCL outperforms sMCL when the target distribution is multimodal, *i.e.*, for extreme values of $x$. However, in the unimodal scenario, the performances of rMCL and sMCL are fairly similar. Furthermore, the 'rMCL*' variant (depicted by crosses) that is lighter regarding memory consumption exhibits performance comparable to the standard rMCL. Additional assessments, which are detailed in Appendix B.2, have demonstrated the robustness of the rMCL when corrupting the training dataset with outliers in the output space.

# 4 Experiments with audio data

## 4.1 Sound source localization as a conditional distribution estimation problem

Sound source localization (SSL) is the task of predicting the successive angular positions of audio sources from a multichannel input audio clip $\boldsymbol{x} \in \mathcal{X}$ [15]. This problem can be cast either as a regression problem (estimating the continuous positions of the sources, *e.g.*, [37]) or a classification one (segmenting the output space into zones in which we wish to detect a source, *e.g.*, [2]). While not requiring the number of sources to be known in advance, the classification approach suffers from the lack of balanced data and limited spatial resolution. On the other hand, the regression approach enables off-grid localization but typically suffers from the source permutation problem [40]. Assuming that the maximum number of overlapping sources $M$ is known, a solution for handling this multi-target regression problem is to predict, at each time frame $t$ depending on the chosen output resolution, a vector accounting for the sources' activity $\boldsymbol{a}_t \in \{0,1\}^M$, as well as azimuth and elevation angles $\boldsymbol{\phi}_t \in \mathbb{R}^M$ and $\boldsymbol{\vartheta}_t \in \mathbb{R}^M$. A model can be trained with a permutation invariant training (PIT) approach [1, 37, 45] using an optimization criterion of the form

$$\mathcal{L}(\theta) = \sum_t \min_{\sigma \in \mathcal{S}_M} \ell_{\text{CE}} \left( \sigma(\hat{\boldsymbol{a}}_t), \boldsymbol{a}_t \right) + \ell_g \left( (\sigma(\hat{\boldsymbol{\phi}}_t), \sigma(\hat{\boldsymbol{\vartheta}}_t)), (\boldsymbol{\phi}_t, \boldsymbol{\vartheta}_t) \right), \tag{13}$$

where $\ell_{\text{CE}}$ and $\ell_g$ correspond respectively to a cross-entropy term and a geometrical loss, the latter being computed only for active sources indexes. $\mathcal{S}_M$ is the set of permutations of $M$ elements and the notation $\sigma(\boldsymbol{z})$ stands for the $M$-dim vector $\boldsymbol{z}$ with its components permuted according to $\sigma$. In the following, we will denote $\boldsymbol{Y}_t$ the set of source positions at time $t$.

With the distribution learning mindset, this task can be seen as an attempt to estimate, at each time step, the sound source position distribution $p(\boldsymbol{y} \mid \boldsymbol{x})$, which can be viewed as a Dirac mixture, $p(\boldsymbol{y} \mid \boldsymbol{x}) \propto \sum_{\boldsymbol{y}_i \in \boldsymbol{Y}_t} \delta_{\boldsymbol{y}_i}(\boldsymbol{y})$, if we suppose that the targets are point-wise, with another Dirac mixture representing the predicted active modes at predicted positions. Therefore, a natural way to evaluate such SSL regression models is to solve the linear assignment problem, *e.g.*, using Hungarian matching with spherical distance as an underlying metric. To handle more general distributions, we propose to generalize the metric used to the Earth Mover's Distance (see Sec. 4.2).

The rMCL framework is well suited to SSL as it allows one to benefit from both the advantages of the regression and classification viewpoints in the same spirit as [41]. There is no need for prior knowledge of the number of sources, and it avoids challenges related to imbalanced spatial positions and the source permutation problem of (13). This method comes at a low computational cost regarding added parameters when opting for a low-level representation shared by the hypotheses and scores heads (see Sec. 3.2). Furthermore, it allows for producing a heat map of the sound sources' positions with a probabilistic prediction, which could otherwise account for their spatial dispersion depending on the chosen law $\pi_k$ selected in the Voronoi cells. Modeling the sound sources as point sources, a delta law will be selected following the formulation of Proposition 1.

## 4.2 Experimental setup

**Datasets.** We conducted our experiments on increasingly complex SSL datasets originally introduced by Adavanne et al. [1]: i) ANSYN, derived from DCASE 2016 Sound Event Detection challenge with spatially localized events in anechoic conditions and ii) RESYN, a variant of ANSYN in reverberant conditions. Each dataset is divided into three sub-datasets depending on the maximum number of overlapping events (1, 2, or 3, denoted as D1, D2, and D3). These sub-datasets are further divided into three splits (including one for testing). Each training split contains 300 recordings (30 s), of which 60 are reserved for validation. Preprocessing is detailed in Appendix A.2. For each experiment, we used all training splits from D1, D2, and D3. Evaluation was then conducted for each test set based on the overlapping levels. Moreover, we present results from additional SSL datasets, including REAL [1] and DCASE19 [3], in Appendix B.3.

**Metrics.** To assess the performance of the different methods, we employed the following metrics:

- The *'Oracle' error* ($\downarrow$): $\mathscr{O}(\boldsymbol{x}_n, \boldsymbol{Y}_n) = \frac{1}{|\boldsymbol{Y}_n|} \sum_{\boldsymbol{y}_m \in \boldsymbol{Y}_n} \min_{k \in [\![1,K]\!]} d \left( f_\theta^k(\boldsymbol{x}_n), \boldsymbol{y}_m \right)$.

- The *Earth Mover's Distance* (EMD $\downarrow$): also known as the 1st Wasserstein distance, it is a distance measure between two probability distributions. In this context, these

are the predicted distribution, $\hat{p}(\boldsymbol{y}\,|\,\boldsymbol{x}_n) = \sum_{k=1}^{K} \gamma_\theta^k(\boldsymbol{x}_n)\delta_{f_\theta^k(\boldsymbol{x}_n)}(\boldsymbol{y})$, and the ground-truth one, $p(\boldsymbol{y}\,|\,\boldsymbol{x}_n) = \frac{1}{|Y_n|}\sum_{\boldsymbol{y}_m \in Y_n}\delta_{\boldsymbol{y}_m}(\boldsymbol{y})$. The EMD solves the optimization problem $W_1\left(p(\cdot|\boldsymbol{x}_n),\hat{p}(\cdot|\boldsymbol{x}_n)\right) = \min_{\boldsymbol{\psi} \in \Psi}\sum_{k=1}^{K}\sum_{\boldsymbol{y}_m \in Y_n}\psi_{k,m}d(f_\theta^k(\boldsymbol{x}_n),\boldsymbol{y}_m)$, where $\boldsymbol{\psi} = \left(\psi(f_\theta^k(\boldsymbol{x}_n),\boldsymbol{y}_m)\right)_{k,m}$ is a transport plan and $\Psi$ is the set of all valid transport plans [21].

While the oracle error assesses the quality of the *best* hypothesis predicted for each target, the EMD metric looks at *all* hypotheses. It provides insight into the overall consistency of the predicted distribution. The EMD also penalizes the overconfidence issue of the WTA with sMCL as described in Sec. 3.4. In order to fit the directional nature of the sound source localization task, these metrics are computed in a spherical space equipped with distance $d(\hat{\boldsymbol{y}},\boldsymbol{y}) = \arccos[\sin(\hat{\vartheta})\sin(\vartheta) + \cos(\hat{\vartheta})\cos(\vartheta)\cos(\phi - \hat{\phi})]$, where $(\hat{\phi},\hat{\vartheta})$ and $(\phi,\vartheta)$ are source positions $\hat{\boldsymbol{y}}$ and $\boldsymbol{y}$, respectively, expressed as azimuth and elevation angles.

**Neural network backbone.** We employed the CRNN architecture of SeldNet [1] as a backbone, which we modified to enable multi-hypothesis predictions. This adaptation involves adjusting the output format and duplicating the last fully-connected layers for hypothesis and scoring heads. As only the prediction heads are adjusted, the number of parameters added to the architecture is negligible. Refer to Appendix A.2 for additional architecture and training details.

**Baselines.** We compared the proposed rMCL with several baselines, each utilizing the same feature-extractor backbone as previously described. The baselines include a Permutation Invariant Training variant ('PIT variant') proposed in [37] for SSL, the conventional WTA setup ('WTA, 5 hyp.') and its single hypothesis variant ('WTA, 1 hyp.'), its $\varepsilon$-relaxed version with $\varepsilon = 0.5$ ('$\varepsilon$-WTA, 5 hyp.') and its top-$n$ variant with $n = 3$ ('top-$n$-WTA, 5 hyp.'), as well as independent ensembles ('IE'). To ensure a fair comparison, we used the same number of 5 hypotheses for the WTAs, sufficient to cover the maximum of 3 sound sources in the dataset (refer to Sec. 4.5 and the sensitivity study in Table 4). For the single hypothesis WTA, the head is updated by only considering the best target, as this fares better than using one update per target. IE was constructed from five such single hypothesis WTA models trained independently with random initialization.

### 4.3  Comparative evaluation of rMCL in SSL

Table 1: **Source localization in anechoic conditions.** Average scores ($\pm$ standard deviation) on ANSYN dataset for PIT, IE, various WTA-based methods, and proposed rMCL. The 'EMD' evaluates all the hypotheses jointly, while the 'Oracle' error only looks at the best hypotheses. D1 corresponds to the single-target setting, while D2 and D3 consider up to 2 and 3 targets, respectively.

| Dataset: **ANSYN** | EMD D1 | EMD D2 | EMD D3 | Oracle D1 | Oracle D2 | Oracle D3 |
|---|---|---|---|---|---|---|
| PIT variant | $6.22 \pm 0.80$ | $14.65 \pm 1.22$ | $23.41 \pm 1.39$ | $3.58 \pm 0.46$ | $10.58 \pm 0.91$ | $18.10 \pm 1.04$ |
| IE (5 members) | $4.05 \pm 0.48$ | $21.64 \pm 2.29$ | $34.34 \pm 2.37$ | $\mathbf{1.24 \pm 0.24}$ | $16.91 \pm 2.05$ | $28.82 \pm 2.25$ |
| WTA, 1 hyp. | $\mathbf{3.97 \pm 0.55}$ | $24.69 \pm 2.72$ | $39.66 \pm 2.67$ | $3.97 \pm 0.55$ | $24.69 \pm 2.72$ | $39.66 \pm 2.67$ |
| WTA, 5 hyp. | $48.22 \pm 1.78$ | $44.41 \pm 1.25$ | $41.83 \pm 0.96$ | $3.56 \pm 0.39$ | $\mathbf{6.53 \pm 0.44}$ | $\mathbf{10.44 \pm 0.58}$ |
| top-$n$-WTA, 5 hyp. | $5.14 \pm 0.80$ | $18.09 \pm 1.32$ | $25.12 \pm 1.30$ | $3.33 \pm 0.46$ | $7.48 \pm 0.79$ | $13.54 \pm 0.96$ |
| $\varepsilon$-WTA, 5 hyp. | $5.51 \pm 0.66$ | $19.20 \pm 1.78$ | $28.39 \pm 1.68$ | $3.62 \pm 0.57$ | $10.86 \pm 1.26$ | $17.44 \pm 1.22$ |
| rMCL, 5 hyp. | $7.04 \pm 0.58$ | $\mathbf{13.87 \pm 0.99}$ | $\mathbf{20.76 \pm 1.04}$ | $3.85 \pm 0.46$ | $7.16 \pm 0.67$ | $11.29 \pm 0.78$ |

Table 2: **Source localization in reverberant conditions**. Results on RESYN dataset, with same table layout as in Table 1.

| Dataset: **RESYN** | EMD D1 | EMD D2 | EMD D3 | Oracle D1 | Oracle D2 | Oracle D3 |
|---|---|---|---|---|---|---|
| PIT variant | $10.53 \pm 0.90$ | $25.09 \pm 2.14$ | $35.05 \pm 1.98$ | $4.89 \pm 0.55$ | $15.92 \pm 1.36$ | $24.95 \pm 1.62$ |
| IE (5 members) | $\mathbf{8.24 \pm 1.02}$ | $26.65 \pm 2.49$ | $38.7 \pm 2.62$ | $\mathbf{3.77 \pm 0.61}$ | $20.95 \pm 2.27$ | $32.85 \pm 2.48$ |
| WTA, 1 hyp. | $8.32 \pm 1.28$ | $29.26 \pm 2.85$ | $43.25 \pm 2.94$ | $8.32 \pm 1.28$ | $29.26 \pm 2.85$ | $43.25 \pm 2.94$ |
| WTA, 5 hyp. | $57.88 \pm 1.71$ | $51.74 \pm 1.36$ | $47.38 \pm 1.26$ | $5.81 \pm 0.58$ | $\mathbf{9.46 \pm 0.71}$ | $\mathbf{13.33 \pm 0.69}$ |
| top-$n$-WTA, 5 hyp. | $42.74 \pm 2.86$ | $37.25 \pm 1.88$ | $36.48 \pm 1.40$ | $6.21 \pm 0.79$ | $11.02 \pm 1.00$ | $17.32 \pm 1.11$ |
| $\varepsilon$-WTA, 5 hyp. | $8.84 \pm 1.09$ | $27.3 \pm 2.52$ | $38.43 \pm 2.42$ | $6.48 \pm 0.95$ | $20.54 \pm 2.33$ | $30.18 \pm 2.15$ |
| rMCL, 5 hyp. | $12.14 \pm 1.12$ | $\mathbf{24.45 \pm 1.91}$ | $\mathbf{32.28 \pm 1.85}$ | $5.74 \pm 0.66$ | $10.5 \pm 0.87$ | $14.6 \pm 0.87$ |

We present in Tables 1 and 2 the results of rMCL and baselines in the anechoic (ANSYN) and reverberant (RESYN) conditions, respectively. The consistency of the results should be noted. For each model, the metrics tend to improve as the number of sources decreases and, under similar conditions, results are marginally lower for more challenging datasets. Unsurprisingly, the single hypothesis approach exhibits strong performances in the unimodal cases (D1), but performs significantly worse in the multimodal cases (D2 and D3). Ensembling (IE) improves its performance but still displays the lack of diversity already exposed in Section 3.4. As for the 5-hypothesis WTA, in its original form, it is ill-suited for multi-target regression due to the overconfidence issue: despite a strong oracle metric, the EMD results are very poor. On the other hand, rMCL surpasses its competitors and shows the best EMD in every multimodal setting. It also consistently obtains the second-best oracle error, only slightly above WTA, while not suffering from overconfidence.

We also present results for REAL and DCASE19 in Appendix B.3, that display similar trends. We did not observe the collapse issue (see Section 3.1) in our settings, neither with WTA nor with the proposed rMCL model. We believe it is solved in practice by the variability of the data samples in the stochastic optimization during training [19]; refer to Appendix B.1 for further discussions.

## 4.4 Combining rMCL with other WTA variants

Table 3: **Combining proposed rMCL with other WTA variants**. Average scores ($\pm$ standard deviation) for source localization in anechoic (top) and reverberant (bottom) conditions.

| Dataset: **ANSYN** | EMD D1 | EMD D2 | EMD D3 | Oracle D1 | Oracle D2 | Oracle D3 |
|---|---|---|---|---|---|---|
| rMCL, 5 hyp. | $7.04 \pm 0.58$ | $13.87 \pm 0.99$ | $20.76 \pm 1.04$ | $3.85 \pm 0.46$ | $\mathbf{7.16 \pm 0.67}$ | $\mathbf{11.29 \pm 0.78}$ |
| top-$n$-rMCL, 5 hyp. | $\mathbf{5.46 \pm 0.62}$ | $13.88 \pm 1.06$ | $21.45 \pm 1.10$ | $4.2 \pm 0.55$ | $8.04 \pm 0.74$ | $13.72 \pm 0.89$ |
| $\varepsilon$-rMCL, 5 hyp. | $5.89 \pm 0.69$ | $\mathbf{12.13 \pm 0.98}$ | $\mathbf{19.95 \pm 1.16}$ | $\mathbf{3.6 \pm 0.54}$ | $8.76 \pm 0.89$ | $14.47 \pm 1.02$ |
| Dataset: **RESYN** | EMD D1 | EMD D2 | EMD D3 | Oracle D1 | Oracle D2 | Oracle D3 |
| rMCL, 5 hyp. | $12.14 \pm 1.12$ | $24.45 \pm 1.91$ | $\mathbf{32.28 \pm 1.85}$ | $\mathbf{5.74 \pm 0.66}$ | $\mathbf{10.5 \pm 0.87}$ | $\mathbf{14.6 \pm 0.87}$ |
| top-$n$-rMCL, 5 hyp. | $9.3 \pm 1.15$ | $23.81 \pm 2.22$ | $33.33 \pm 2.06$ | $6.99 \pm 0.85$ | $13.43 \pm 1.40$ | $19.72 \pm 1.29$ |
| $\varepsilon$-rMCL, 5 hyp. | $\mathbf{8.64 \pm 1.03}$ | $\mathbf{22.82 \pm 2.12}$ | $32.47 \pm 1.97$ | $6.08 \pm 0.91$ | $18.39 \pm 2.07$ | $26.92 \pm 1.94$ |

As noted in Section 3.2, rMCL's improvements are, in theory, orthogonal to that of other WTA variants. In this section, we combine the rMCL with top-$n$-WTA and $\varepsilon$-WTA approaches into respectively top-$n$-rMCL and $\varepsilon$-rMCL. The results are shown in Tables 3. We note that while top-$n$ rMCL does not provide significant improvement to rMCL, $\varepsilon$-rMCL, on the other hand improves EMD scores at the expense of increased oracle error. We observe similar effects when this method is applied to WTA, indicating that there are indeed some additive effects of $\varepsilon$-WTA and rMCL. Also note that with regards to Tables 1 and 2, all the proposed variants still get better EMD than all competitors under multimodal conditions, showing the robustness of our method.

## 4.5 Effect of the number of hypotheses

The impact of varying the number of hypotheses on the performance of the rMCL model is presented in Table 4 and Figure B.4. First and foremost, we notice the anticipated trend of the oracle metric improving as the number of hypotheses increases. Concerning the EMD metric, the single hypothesis model, which avoids errors from negative hypotheses, is most effective in handling unimodal

Table 4: **Sensitivity analysis**. Effect of the number of hypotheses on the performance of rMCL.

| Dataset: **ANSYN** | EMD D1 | EMD D2 | EMD D3 | Oracle D1 | Oracle D2 | Oracle D3 |
|---|---|---|---|---|---|---|
| PIT variant | $6.22 \pm 0.80$ | $14.65 \pm 1.22$ | $23.41 \pm 1.39$ | $3.58 \pm 0.46$ | $10.58 \pm 0.91$ | $18.10 \pm 1.04$ |
| WTA, 1 hyp. | $\mathbf{3.97 \pm 0.55}$ | $24.69 \pm 2.72$ | $39.66 \pm 2.67$ | $3.97 \pm 0.55$ | $24.69 \pm 2.72$ | $39.66 \pm 2.67$ |
| rMCL, 3 hyp. | $9.89 \pm 0.95$ | $14.37 \pm 0.91$ | $20.96 \pm 1.03$ | $5.65 \pm 0.73$ | $8.6 \pm 0.63$ | $13.42 \pm 0.86$ |
| rMCL, 5 hyp. | $7.04 \pm 0.58$ | $\mathbf{13.87 \pm 0.99}$ | $\mathbf{20.76 \pm 1.04}$ | $3.85 \pm 0.46$ | $7.16 \pm 0.67$ | $11.29 \pm 0.78$ |
| rMCL, 10 hyp. | $9.14 \pm 0.76$ | $15.2 \pm 0.84$ | $21.28 \pm 0.96$ | $2.94 \pm 0.35$ | $4.76 \pm 0.39$ | $7.54 \pm 0.50$ |
| rMCL, 20 hyp. | $9.13 \pm 0.71$ | $16.04 \pm 0.84$ | $22.55 \pm 0.87$ | $\mathbf{2.06 \pm 0.22}$ | $\mathbf{3.61 \pm 0.30}$ | $\mathbf{5.83 \pm 0.37}$ |

distributions. In the case of the EMD metric applied to multimodal distributions, we observe that multiple hypothesis models improve the results, but excessively increasing the number of hypotheses may marginally degrade performances. A further study could examine this phenomenon in detail, which may be related to the expressive power of the hypothesis heads or the accumulation of errors in score heads predictions.

## 5   Discussion and limitations

In the audio experiments, the performance of rMCL was found to be affected by the number of hypotheses to tune depending on the complexity of the dataset. Moreover, as with the other variants of WTA, fixing the overconfidence issue with rMCL slightly degrades the performance of the best hypothesis (oracle error) for a reason that is yet to be determined. Otherwise, while $\varepsilon$-WTA behaves as expected with rMCL, trading off the quality of the best hypotheses for overall performance, top-$n$-WTA does not exhibit the same behavior. This discrepancy warrants further investigation.

The probabilistic interpretation of rMCL, as presented in Algorithm 1 and in Sec. 3.3, states that the different hypotheses would be organized in an optimal partition of the output space, forming a Voronoi tessellation. Ideally, each hypothesis would capture a region of the distribution, and the scores representing how likely this zone would activate in a given context. This interpretation remains theoretically valid whenever train and test examples are sampled from the same joint distribution $p(\boldsymbol{x}, \boldsymbol{y})$. However, in realistic settings, the hypotheses might not adhere to meaningful regions, but this could be controlled and evaluated, provided that an external validation dataset is available. In future work, we intend to use calibration techniques [16, 39] to identify and alleviate these issues.

To sum up, this paper proposes a new Multiple Choice Learning variant, suitable for tackling the MCL overconfidence problem in regression settings. Our method is based on a learned scoring scheme that handles situations where a set of targets is available for each input. Furthermore, we propose a probabilistic interpretation of the model, and we illustrate its relevance with an evaluation on synthetic data. Its practical usefulness is also demonstrated in the context of point-wise sound source localization. Further work could include a specific study about the specialization of the predictors, and the validation of the proposed algorithm in increasingly complex real-world datasets.

## Acknowledgments

This work was funded by the French Association for Technological Research (ANRT CIFRE contract 2022-1854). We would like to thank the reviewers for their valuable feedback.

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
