# OpenReview forum: "Resilient Multiple Choice Learning: A learned scoring scheme with application to audio scene analysis"
_NeurIPS.cc/2023/Conference — NeurIPS 2023 poster_

### Official Review · Reviewer_3WKf · 2023-07-10

**Soundness:** 3 good
**Presentation:** 2 fair
**Contribution:** 3 good
**Rating:** 6
**Confidence:** 1

**Summary:**

The paper tackles the problem of multiple-choice learning (MCL) in the regression setting with specific focus on the overconfidence problem and hypothesis collapse problem found in previous approaches for MCL where predictions from the heads corresponding to rare events are overestimated. The proposed rMCL model frames the problem as a multimodal conditional distribution estimation problem. The authors propose a new loss function that adds a new hypothesis scoring loss to the existing multi-target winner takes all (WTA) loss.

The proposed algorithm is compared against other baseline models on both a toy dataset, as well as a sound source localization (SSL) task. The experiments on the toy datasets visually show the effectiveness of the rMCL approach in tackling the overconfidence problem as compared to a method that does not use the additional loss term. The results on the SSL dataset show that the rMCL algorithm is practically applicable to problems with multimodal outputs with consistently better performance in the case of multiple sound sources as compared to the other algorithms.

**Strengths:**

- The presented approach builds on top of existing literature and solves the overconfidence problem using an additional term to the existing loss function.
- The authors visually demonstrate the effectiveness of the approach using a simple toy dataset, and follow up with application to a real-world regression task (SSL).

**Weaknesses:**

- The experiment on SSL is not very clear to me and might benefit from some additional details. Specifically, I am not clear on the interpretation of the EMD and oracle error. My understanding is that for each input sound snippet, the model predicts the angles of each source from each of the output heads. Does the oracle only count the best prediction out of all the source angles predicted, while the EMD accounts for the overall error?

**Questions:**

See above

**Limitations:**

The authors have addressed limitations with the presented work.

---

> ### Author Rebuttal · Authors · 2023-08-10
>
> We would like to thank the reviewer for the remarks and the feedback on the paper.
>
> The reviewer is indeed correct in that the EMD (Earth mover's distance or Wasserstein-1 metric [A]) considers all the hypotheses predicted and their associated scores. When it comes to the oracle metric, it evaluates only the best prediction for each target source (e.g., [11,20,29,24,22,18,6]). To be more detailed, we elaborate here-after on the SSL task, and the metrics related to multiple choice learning. We will clarify their presentation in the paper accordingly.
>
> Given audio tracks recorded from a microphone array, the SSL task consists of predicting, at a given temporal rate, the positions of specific sound sources one is interested in. Sound sources can appear or disappear in the record, and their number can vary. As for the position, in those benchmarks, we are only interested in the angular position.
>
> Because of the uncertain and multi-modal nature of the prediction task, it can be useful to cast it in a multiple choice learning or distribution learning framework. In such settings, the models output different plausible predictions or hypotheses for any given input. The difficulty that arises then is how to assess the quality of the multiple predictions of the model when only one realization of the ground-truth distribution is observed for any given input.
>
> Whenever a single target is available, which corresponds to one source to localize in SSL, one possibility is to compute the error for the prediction closest to the ground truth, that is the Oracle metric: we get 0 oracle error if the ground-truth is in the pool of predictions. When multiple targets are present, the oracle metric averages over the best hypothesis for each target [6]. Heavily used in the MCL framework [11,20,29,24,22,18,6], this metric therefore informs about the mean quality of the best hypotheses predicted.
>
> The other metric, the Earth Mover’s distance (EMD), computes the optimal transport cost [A] between the ground truth and predicted distribution, both being cast as a mixture of Diracs in our case. Indeed in those audio datasets, sound sources can be assumed to be point-wise, and the ground-truth distribution to predict can be cast as a uniform mixture of Diracs. This considers all the hypotheses, weighted by their normalized score, and allows for a more complete evaluation; it informs about the global consistency of the hypotheses predicted [22].
>
> Both the Oracle and the EMD are equipped with an underlying distance adapted to the geometry of the problem. Since we are dealing with angles only, this underlying distance is angular distance here.
>
> [A] Kantorovitch, L. (1958). On the translocation of masses. Management science, 5(1), 1-4.

---

### Official Review · Reviewer_YyHX · 2023-07-12

**Soundness:** 3 good
**Presentation:** 3 good
**Contribution:** 3 good
**Rating:** 6
**Confidence:** 5

**Summary:**

The authors propose Resilient Multiple Choice Learning (rMCL), a modification of the Multiple Choice Learning (MCL) approach, for conditional distribution estimation in regression contexts where each input can have multiple target samples. While MCL is a straightforward strategy for multimodal density estimation, it uses the Winner-Takes-All (WTA) loss for various hypotheses. In regression situations, the prevailing MCL versions focus on combining the hypotheses, which may compromise the diversity of the predictions.
In contrast, rMCL employs a new learned scoring system, which is supported by a mathematical framework based on Voronoi tessellations of the output space. This approach allows for a probabilistic interpretation of the results. The authors tested rMCL using synthetic data and found it to be effective. They also applied it to the problem of sound source localization, demonstrating its practical utility and the relevance of its interpretation.


**Strengths:**

Novel Approach: The proposed Resilient Multiple Choice Learning (rMCL) provides a fresh perspective to tackle the Multiple Choice Learning (MCL) overconfidence problem, especially in regression settings. This offers a new method for researchers and engineers to approach this issue.

Learned Scoring: rMCL is based on a learned scoring scheme that handles multi-target settings. This flexibility could allow rMCL to perform well across a variety of tasks and datasets.

Probabilistic Interpretation: The authors provide a probabilistic interpretation of the model, which could help in understanding the model's behavior, tuning its performance, or extending it to new applications.

Evaluation and Application: The paper demonstrates the practical utility of rMCL, especially in the context of sound source localization. It illustrates the resilience of the model in both synthetic and real-world datasets, showing that the method can handle real-world complexity.

Advantages Over Previous Methods: rMCL, applied to the Sound Source Localization (SSL) problem, seems to alleviate the issues related to imbalanced spatial positions and the source permutation problem. It does not require prior knowledge of the number of sources, which is a significant advantage in practical applications.



**Weaknesses:**

The method heavily relies on the dependence on high-quality data: Like many machine learning models, the performance of rMCL might be significantly impacted by the quality and representativeness of the training data. Further evaluation might be required based on whether the model is affected in the presence of noise in the data and the quality of the labels.

The performance of the rMCL approach may not be suitable when one is approaching to scale the data wherein the size and complexity of the dataset would affect the nature of the machine learning task at hand especially when one considers the hypothesis space which is large.

It's not clear how rMCL would respond to different types of noise in the data or how robust it is to outliers.



**Questions:**

Based on the weakness it would be helpful for authors to provide comments on how the rMCL approach might work in case of noise in the data and whether there is any comparative analysis of WinnerTakesAll type approaches which take into consideration of the label data noise.

Despite the probabilistic interpretation offered by the rMCL approach, it might still be challenging to understand and explain the model's decision-making process, which could be a limitation in certain applications where interpretability is crucial. How does authors see the probabilistic interpretation aiding/modified to overcome the interpretability of the model learnt? It might be the case where the probabilistic interpretation may yield incorrect result with overfitting of the model.

**Limitations:**

Authors do state the limitations of the work clearly.

They review the performance of the model and conclude that it appears that the performance of Resilient Multiple Choice Learning (rMCL) is influenced by the number of hypotheses being considered. Particularly, attempting to predict an overly large number of hypotheses without prior knowledge can introduce errors into score predictions. In addition, using rMCL to address the overconfidence issue inherent in Winner-Takes-All (WTA) variants can slightly degrade the performance of the best hypothesis, though the reason for this is not currently known. Lastly, while ε-WTA seems to behave as expected with rMCL, the same is not observed for top-n-WTA, necessitating further study.

---

> ### Author Rebuttal · Authors · 2023-08-10
>
> We would like to thank the reviewer for their insightful comments.
>
> > Based on the weakness it would be helpful for authors to provide comments on how the rMCL approach might work in case of noise in the data and whether there is any comparative analysis of WinnerTakesAll type approaches that take into consideration of the label data noise.
>
> The reviewer is asking about the robustness of the model in the presence of label noise, which is a challenging problem. The label noise corresponds to the aleatoric uncertainty [8,4], which can not be reduced with more data (e.g., corresponding to measurement error).
>
> To the best of our knowledge, there is no comparative analysis of Winner-Takes-All type approaches in the context of noise; we provide, in the next section, insights about the resilience of the proposed model in the presence of outliers in the data.
>
> **Performance of rMCL in the presence of target outliers**
>
> We thank the reviewer for their question which prompted some interesting investigation. We believe that rMCL, thanks to the partition into Voronoi tessellations and the scoring scheme, has good properties for handling outliers. To illustrate this, we propose the following experiment.
>
> Let's consider a setting where we have outliers in the training dataset, for instance, the toy-use case presented in the paper, where for each training example sampled, the probability of getting an outlier is $p \ll 1$, e.g., modeled with a bivariate Cauchy distribution. Then, whenever an outlier is sampled, one hypothesis will be pushed towards it with its associated score heads updated. As the training goes on, some of the hypotheses will manage the outliers samples; let's name them the "outlier hypotheses". Thanks to the proposed hypothesis scoring heads, the model will also learn the probability that an outlier hypothesis is chosen for a given training sample. Provided that the outlier likelihood is $p\ll 1$, the scoring heads will therefore prevent outlier hypotheses output from deteriorating the quality of the predicted distribution by rMCL. In Fig.A an illustration of this phenomenon is proposed using a Cauchy distribution (we used $p=0.02$). We notice the above-explained phenomenon, where the so-called outlier hypotheses account for the outlier samples, while the other hypotheses lie in the square $[-1,1]^{2}$ representing the samples from the ground-truth distribution.
>
> Provided that the probability of sampling an outlier $p$ is small enough and the outliers are far enough from the ground-truth distribution to predict, the proposed rMCL model is therefore potentially robust to outliers. In this case, some specific hypotheses, namely the outlier hypotheses, will be assigned to them, preventing the non-outliers hypothesis from being heavily affected. At inference time, it will indeed be possible to set to zero the very low-score hypotheses given an arbitrary threshold, so that the outlier hypotheses are not taken into account.
>
> > "Despite the probabilistic interpretation offered by the rMCL approach, it might still be challenging to understand and explain the model's decision-making process, which could be a limitation in certain applications where interpretability is crucial. How do authors see the probabilistic interpretation aiding/modified to overcome the interpretability of the model learnt? It might be the case where the probabilistic interpretation may yield incorrect results with overfitting of the model."
>
> The probabilistic interpretation allows us to state that the different hypotheses would be organized in an optimal partition of the output space forming a Voronoi tessellation, providing insights about the distribution to predict through the score heads. In an ideal case, this would lead to each hypothesis capturing a region of the distribution, and the scores being how likely this zone would activate in a given context. Of course, in a realistic setting, the hypothesis might not adhere to a meaningful region, but this can be controlled and evaluated provided we have additional annotations in our data.
>
> Another asset of our approach is that the scoring function, similarly to a classifier, should provide an insight about the confidence of our model on each hypothesis. While the predicted probabilities can be wrong as stated by the reviewer, we can evaluate and calibrate the scoring function independently on a validation set [A,B], which should allow us to identify and alleviate such issues.
>
> We hope that we understood the concerns of the reviewer correctly and that these considerations address them. We would be happy to further discuss these topics and clarify any potential misunderstanding.
>
> [A] Guo, C., Pleiss, G., Sun, Y., & Weinberger, K. Q. (2017, July). On calibration of modern neural networks. In International conference on machine learning (pp. 1321-1330). PMLR.
>
> [B] Song, H., Diethe, T., Kull, M., & Flach, P. (2019, May). Distribution calibration for regression. In International Conference on Machine Learning (pp. 5897-5906). PMLR.

---

### Official Review · Reviewer_yLZS · 2023-07-27

**Soundness:** 3 good
**Presentation:** 3 good
**Contribution:** 2 fair
**Rating:** 6
**Confidence:** 3

**Summary:**

This paper proposes a technique for resilient Multiple Choice Learning (rMCL), which extends the vanilla Muliple Choice Learning (MCL) paradigm to conditional distributions for regression where multiple targets maybe sampled for each training input. It is known that MCL uses multiple scoring heads to score multiple hypothesis for a given input and suffers from the twin challenges of hypothesis collapse (where only a small subset of the possible prediction heads are trained well, as a result of the Winner Take All strategy) and overconfident predictions (where rare classes are overly represented). This work focuses principally on addressing  the latter issue. A key feature of the method is the use of a learned scoring scheme based on Voronoi tessellations which lends itself to a probabilistic interpretation. Results are reported on the sound source localization task and on synthetic data and compared against some standard alternatives.

**Strengths:**

The following are the key strengths of the proposed approach:
1. This is possibly the first work that extends MCL to a regression setting.
2. The proposed solution attempts to overcome the overconfidence problem of a standard MCL system by casting the MCL as a conditional distribution estimation technique while allowing for a probabilistic interpretation of the same.
3. Experiments show that the method, which is capable of assigning prediction probabilities to low density regions, is also interpretable.
4. The code for the proposed method has been made available.

**Weaknesses:**

The following are some of the principal weaknesses of the proposed approach:
1. The proposed method does not deal with the hypothesis collapse problem of MCL methods.
2. The contrast with prior approaches is not well brought out in the Related Works section.
3. No performance is reported on more recent sound source localization datasets such as LOCATA or the DCASE 2019 dataset
4. Moreover, the proposed approach falls way short of competing methods on the Oracle metrics, on both the reported datasets. The authors present too much emphasis on just the multimodal EMD results without justifying why the drop in Oracle distance metric should be ignored.

**Questions:**

Following are some of the questions/comments for the authors:
1. How does the current work compare against prior works in Multilabel Learning? [1, 2]
2. How do the methods perform on the Euclidean distance measure, compared with the Oracle Error?
3. This reviewer fails to see the value in ensembling the proposed approach with WTA (Table 3). Why not combine with IE or PIT then?
4. Please rephrase Line 111: "More precisely...."

References:
[1] Zhu, X., Li, J., Ren, J., Wang, J. and Wang, G., 2023. Dynamic ensemble learning for multi-label classification. Information Sciences, 623, pp.94-111.
[2] Kim, Y., Kim, J.M., Akata, Z. and Lee, J., 2022. Large loss matters in weakly supervised multi-label classification. In Proceedings of the IEEE/CVF Conference on Computer Vision and Pattern Recognition (pp. 14156-14165).

**Limitations:**

Yes the limitations of the method have been well addressed.

---

> ### Author Rebuttal · Authors · 2023-08-10
>
> We thank the reviewer for their feedback and comments, as well as for the suggestions for extending the experimental results of the paper. We provide here a detailed answer to the raised concerns.
>
> **The metrics interpretation**
>
> > “The authors present too much emphasis on just the multimodal EMD results”
>
> We insist on multimodal settings rather than unimodal ones because rMCL is suited for multimodal density estimation. The proposed approach is particularly relevant when the output conditional mean is not the best solution, which generally corresponds to multimodal distributions to predict.
>
> > “without justifying why the drop in Oracle distance metric should be ignored.”
>
> In our experiments, the focus was primarily on the EMD metric, as the Oracle does not address the issue of overconfidence. It indeed only considers the best hypotheses, without accounting for the global consistency of the prediction; if a low probability density zone is overestimated, this phenomenon will not be measured by the Oracle metric. This precision will be added to the paper.
>
> > “How do the methods perform on the Euclidean distance measure, compared with the Oracle Error?”
>
> As highlighted in L.257, the Oracle error is computed using an underlying distance $d$ adapted to the output geometry, e.g., the Euclidean distance in the toy example or the spherical distance in the SSL task. For those benchmarks, we are only interested in the angular instead of Cartesian positions, therefore computing an Euclidean distance is not representative of the task. This point will be clarified in the paper.
>
> > “This reviewer fails to see the value in ensembling the proposed approach with WTA (Table 3). Why not combine with IE or PIT then?”
>
> These choices are explained by the fact that our approach can be seen as an extension of the WTA training scheme. As such we wanted to evaluate how our approach would combine with commonly used WTA extensions (top-$n$ and $\varepsilon$). It is not straightforward to combine it with PIT or IE. Indeed PIT is not related to MCL and IE was constructed from single-hypothesis WTA models that are not amenable to our method.
>
> **Performance on more SSL datasets**
>
> Following the reviewer's suggestion, we conducted more experiments on SSL datasets: REAL [1] and DCASE 2019 [A], where the maximum overlapping events are three and two. As with the manuscript's tables, model evaluation occurred separately in unimodal and multimodal conditions. For REAL and DCASE19, we computed mean metric values after 2 and 3 training runs, respectively, presented in Tables A and B.
>
> **Tables A & B reveal trends aligned with the paper's findings:**
>
> - Consistent with Section 4.5's analysis, increasing the number of hypotheses improves the oracle, but also degrades slightly the EMD while still surpassing the PIT baseline in multimodal settings.
> - The winner-takes-all approach and the IE variant, with one hypothesis and a single target update (see L.21-23 in Supplementary material), still outperform the other methods whenever a single source position is to be predicted.
> - While vanilla WTA slightly outperforms our method on the Oracle metric with the same number of hypotheses, a significant gap remains in EMD metric when predicting multiple hypotheses (e.g., 5).
> - rMCL shows consistent performance across datasets, where top-n and $\varepsilon$-WTA can show a very wide disparity.
>
> The optimal number of hypotheses on DCASE19 is lower than in previous datasets, probably due to its less multimodal nature. We conducted various visualizations to compare baselines and the proposed approach, confirming the competitive performance on those two SSL datasets.
>
> **The collapse problem**
>
> In our audio experiments, neither the vanilla WTA nor the proposed rMCL model exhibited collapse. We confirmed this by analyzing histograms of winner hypothesis heads from trained models during testing, as illustrated by Fig. A's 20-hypothesis rMCL model trained on ANSYN. As mentioned in [13, p.8], we think collapse is in practice solved by the variability of the data samples and the training stochasticity. This is why we did not study the collapse problem, but we will include this discussion in the Supplementary.
>
> **Comparison with prior work in Multi-label learning [A,B]**
>
> We thank the reviewer for the suggested references, to be included in the Related Work. [A] approaches the topic of Multi-label learning from the ensemble learning perspective, while [B] addresses multi-label classification in images with missing labels, presenting a variant of multiple choice learning. Each hypothesis in this context would aim to predict a possible class present in the image while expecting diverse predictions [18,20,29]. These two approaches are, however, classification methods whereas our paper focuses specifically on MCL for regression tasks. Adapting such methods to multi-source regression is not straightforward.
>
> **About the related work section**
>
> The related work section will be enhanced, focusing on contrasting it with the paper's contribution.
> - The link between uncertainty estimation and MCL will be made clearer through ensembling [20] (L.59-60 in the manuscript).
> - The contrast with previous works in MCL [13,20,29,18,11,24,22,6] will be emphasized; ours tackles the overconfidence problem in regression settings without merging the hypotheses (e.g., [13,24,22]), by revisiting [29]. This results in a gain in diversity of the hypotheses and is suitable for extending the MCL probabilistic interpretation proposed in [24].
>
> As suggested, the sentence in L.111 will be rephrased.
>
> [A] Zhu, X., Li, J., Ren, J., Wang, J., & Wang, G. (2023). Dynamic ensemble learning for multi-label classification. Information Sciences, 623, pp. 94-111.
>
> [B] Kim, Y., Kim, J. M., Akata, Z., & Lee, J. (2022). Large loss matters in weakly supervised multi-label classification. In Proceedings of the IEEE/CVF Conference on Computer Vision and Pattern Recognition (pp. 14156-14165).

---

> > ### Comment · Reviewer_yLZS · 2023-08-15
> > **Thanks for addressing my concerns**
> >
> > In light of the additional experimental results presented and the response to my other questions, I am raising my score.

---

### Author Rebuttal · Authors · 2023-08-10

We would like to thank the reviewers for their remarks and suggestions, which will allow us to improve the quality of the paper.

We summarize here the main changes that will be made to the submission in a next revision, in accordance with the reviewers inputs. Please refer to the individual responses for more detailed comments about those changes, as well as answers to the questions.

- We provide results on two additional sound source localization (SSL) datasets; REAL [1] and DCASE19 [A] in Tables A and B of the rebuttal. The manuscript will be updated accordingly [Reviewer yLZS].
- We will add new insights about the resilience of the proposed rMCL model in the presence of label noise and outliers (See Figure B of rebuttal) in the Supplementary [Reviewer YyHX].
- We will also expand on the motivations for the probabilistic interpretation and include the discussion in the main paper [Reviewer YyHX].
- A discussion regarding the collapse problem will be added in the Supplementary material [Reviewer yLZS].
- The interpretations of the metrics used, the EMD and Oracle will be clarified in the paper [Reviewers YyHX and 3WKf].
- Related work will be improved [Reviewer yLZS].

[A] Adavanne, S., Politis, A., & Virtanen, T. (2019). A multi-room reverberant dataset for sound event localization and detection. arXiv preprint arXiv:1905.08546.

---

### Decision · Program_Chairs · 2023-09-21

**Decision:**

Accept (poster)

**Comment:**

This paper proposes a novel scoring scheme for Multiple Choice Learning (rMCL) in the multiple-target regression setting. This new scoring scheme is supported by casting the output space in Voronoi tessellations, overcoming the need to merge hypotheses (which sacrifices the diversity in predictions) and allows for a probabilistic interpretation.

All of the scores are positive. MCL often suffers from two challenges, hypothesis collapse and overconfident predictions. All reviewers felt the paper’s synthetic experiments demonstrated how the new method overcomes the overconfident predictions. One reviewer felt the collapse problem wasn’t addressed, and the authors illustrated in the rebuttal that the problem was not a concern in the application of sound source localization (SSL). In response to reviewers request, the authors also conducted new experiments on two more recent SSL datasets and found that the results aligned with the paper’s findings.

Initially, there was concern that the proposed method may not work well in the presence of noise and outliers. In the rebuttal the authors gave a detailed response on how prior work approached different kinds of noise and how their method is able to handle outliers. The authors also provided additional explanations on how their method offers interpretability. Reviewers found the insights and proposed experiments valuable.

Overall, reviewers found the work novel, technically sound, and practically applicable. Hence the recommendation is to accept the paper.